# Podoplanin Expression in Early-Stage Colorectal Cancer-Associated Fibroblasts and Its Utility as a Diagnostic Marker for Colorectal Lesions

**DOI:** 10.3390/cells13201682

**Published:** 2024-10-11

**Authors:** Shuichi Tsukamoto, Takayuki Kodama, Mari Nishio, Manabu Shigeoka, Tomoo Itoh, Hiroshi Yokozaki, Yu-ichiro Koma

**Affiliations:** 1Division of Pathology, Department of Pathology, Kobe University Graduate School of Medicine, Kobe 650-0017, Japan; takodama1991@yahoo.co.jp (T.K.); marin@med.kobe-u.ac.jp (M.N.); mshige@med.kobe-u.ac.jp (M.S.); hyoko@med.kobe-u.ac.jp (H.Y.); koma@med.kobe-u.ac.jp (Y.-i.K.); 2Division of Diagnostic Pathology, Department of Pathology, Kobe University Graduate School of Medicine, Kobe 650-0017, Japan; tomitoh@med.kobe-u.ac.jp

**Keywords:** podoplanin, cancer-associated fibroblasts, colorectal carcinoma, diagnostic marker, tumor microenvironment

## Abstract

(Background) Cancer-associated fibroblasts (CAFs) are major cancer stromal components. CAFs have diverse functions and cell origins. Podoplanin (PDPN), a lymphatic vessel marker, is also a CAF marker in certain cancers. On daily diagnosis of early colorectal carcinoma (CRC), PDPN upregulation in the stroma is often encountered, suggesting PDPN-positive CAFs have emerged. However, PDPN-positive CAFs in early CRC have not been studied well. (Methods) On immunohistochemistry, PDPN expression in the lamina propria or stroma of adenomas, early CRCs, and neuroendocrine tumors, their normal neighbors, and non-neoplastic colorectal lesions were compared. Single-cell RNA sequencing (scRNA-seq) of CRC was used to explore *PDPN*^high^ CAFs’ cell origins. (Results) Reticular cells or pericryptal fibroblasts in the lamina propria of adenomas and early CRCs showed higher PDPN expression than did normal mucosae and non-neoplastic lesions (*p* < 0.01). Pericryptal PDPN expression was a diagnostic feature of adenomas and early CRCs. scRNA-seq of CRCs highlighted that *PDPN*^high^ CAFs had distinctly higher *COL4A1*, *COL4A2*, and *WNT5A* expression, unlike well-known CAFs characterized by high *FAP, POSTN*, or *ACTA2* expression. (Conclusions) We demonstrated that pericryptal fibroblasts and reticular cells in the lamina propria are origins of early-stage CRC CAFs and thus have potential as a diagnostic marker for distinguishing colorectal non-neoplastic from neoplastic lesions.

## 1. Introduction

Cancer-associated fibroblasts (CAFs) are major constituents of cancerous tissues or cancer microenvironments. They contribute to cancer progression by producing various growth or angiogenic factors and a fibrous stroma to protect cancer cells from immune cells or anticancer drugs, including cytotoxic agents and immune checkpoint inhibitors [1,2]. Targeting “malignant” CAFs shows promise in cancer treatment. However, CAFs are highly heterogeneous in function (e.g., inflammatory or myofibroblastic CAFs [3]) and origin (e.g., derived from resident fibroblasts [4], mesenchymal stem cells [5], or adipocytes [6]), posing challenges for CAF-targeted therapy. Reflecting this complexity, numerous CAF markers, such as fibroblast activation protein (FAP), alpha-smooth muscle actin (αSMA), or periostin (POSTN), have been identified [3,7,8].

Podoplanin (PDPN) is a small transmembrane glycoprotein expressed in various cell types, including lymphatic endothelial cells (LECs) and fibroblastic reticular cells (RCs). In the normal state, PDPN endows LECs and RCs with contractility via RhoA/RhoC signaling. During the inflammatory process, C-type lectin-like receptor 2 on dendritic cells binds to PDPN on RCs, weakening their contractility. This interaction relaxes the reticular network, facilitating immune cell migration from inflamed sites to regional lymph nodes [9]. Therefore, PDPN acts as a regulator of immune processes.

In the context of diagnostic pathology, PDPN is a well-known marker for lymphatic vessels detected by immunohistochemistry (IHC) using the D2-40 antibody. It is used to assess lymphatic invasion (LI) of tumor cells. PDPN is also expressed in some CAFs. PDPN-positive CAFs (podCAFs) are associated with poor prognosis in pancreatic [10] and lung adenocarcinomas (AC) [11] or with good prognosis in advanced colorectal carcinoma (CRC) [12]. Occasionally, cancer cells surrounded by podCAFs can be mistaken for LI. Therefore, pathologists should be aware of the different PDPN expression patterns of CAFs with reference to those of lymphatic vessels to avoid diagnostic errors. When diagnosing early CRCs suitable for endoscopic resection, PDPN upregulation in cancerous regions is often observed. For example, Nakayama et al. previously reported that 78% (50/65) of adenomas, 94% (30/32) of adenocarcinomas in situ, and 100% (8/8) of submucosally invasive adenocarcinomas had pericryptal PDPN expression [13]. However, PDPN-positive stromal cells in early CRCs have not been studied well from the viewpoint of CAF biology. Here, we investigated the significance of PDPN as a potential CAF marker in CRC and proposed the diagnostic value of stromal PDPN expression in distinguishing between colorectal non-neoplastic and neoplastic lesions.

## 2. Materials and Methods

### 2.1. Tissue Samples

Here, we classified CRC into two steps: early AC, which includes (i.e., intramucosal AC (high-grade dysplasia/adenoma according to the World Health Organization [WHO] classification [14]) or AC invading up to the submucosa, and advanced AC, defined as AC invading the muscularis propria or deeper. We randomly and retrospectively selected five types of colorectal lesions: early AC, adenoma, neuroendocrine tumor (NET), non-specific inflammation, and mucosal prolapse syndrome (MPS), which were resected or biopsied endoscopically. Early ACs and adenomas were diagnosed according to the 9th edition of the Japanese Classification of Colorectal, Appendiceal, and Anal Carcinoma [15]. All tissue samples were obtained at Kobe University Hospital and pathologically diagnosed between January 2020 and July 2023. Non-neoplastic lesion samples (inflammation and MPS) were obtained from patients without known colorectal tumors. Details of collected samples are summarized in Table 1. This study was conducted in accordance with the Declaration of Helsinki and approved by the Institutional Review Board of Kobe University (B230125). All samples were fixed in 10% neutral buffered formalin, embedded in paraffin wax (FFPE), sliced to a thickness of 4 µm, and histologically diagnosed.

### 2.2. Immunohistochemistry and Special Staining

For early AC and NET cases, PDPN IHC (clone D2-40; dilution 1:4; Agilent Technologies, Inc., Santa Clara, CA, USA) was performed using the VENTANA BenchMark ULTRA (Roche, Basel, Switzerland) to evaluate LI as part of the routine diagnostic process.

For adenoma, inflammation, and MPS cases, PDPN IHC (clone D2-40; dilution 1:4; Agilent Technologies, Inc.) was performed using the Leica Bond-MAX (Leica Biosystems, Wetzlar, Germany) with 3,3′-diaminobenzidine (Dako Cytomation, Glostrup, Denmark). Using LECs as an endogenous positive control, it was confirmed that the immunostaining properties of PDPN did not vary between the two IHC methods. The intensity of PDPN expression was classified as high, low, or negative by comparing it with that of LECs: high, equal to, or stronger than LECs; low, weaker than LECs; and negative, no expression. Pericryptal PDPN expression was defined as positive when continuous PDPN expression surrounded ≥50% of glands [16]. We calculated “pericryptal positivity” as the percentage number of PDPN-positive glands among all glands of interest. PDPN immunoreactivity in the lamina propria (LP) and tumor stroma was determined by observation and agreement among three pathologists (S.T., T.K., and Y.-i.K.). For pericryptal positivity, S.T. and T.K. independently calculated percentages for each case, and then percentages <20% of gap were averaged. For the cases with ≥20% gap in the positivity score, S.T. and T.K. reviewed the cases together and decided the positive or negative scores for controversial glands again. Thereafter, the score by T + K was calculated. The gaps in the score by T + K and by Y.-i.K. were averaged or corrected in the same way, and final pericryptal positivities were determined. p53 IHC (clone DO-7, 1:2000, Agilent Technologies, Inc.) was performed for 67 early AC patients with available FFPE blocks and for all samples of the adenoma, inflammation, and MPS using the Leica Bond-MAX (Leica Biosystems) with 3,3′-diaminobenzidine (Dako Cytomation). p53 expression pattern was classified into aberrant (bulky overexpression, complete loss, or cytoplasmic expression) and wild-type (not aberrant) [17].

For selected early AC cases, αSMA IHC (clone 1A4; no dilution; Agilent Technologies) using the VENTANA BenchMark ULTRA (Roche) and reticulin stain were performed.

### 2.3. Immunofluorescence

For two early AC cases, immunofluorescence using FFPE tissue samples was performed. Antibodies against PDPN (D2-40, as used in IHC) and αSMA (ab5694, 1:100, Abcam, Cambridge, UK) were employed for the primary reaction using Leica Bond-MAX. After washing three times with phosphate-buffered saline (PBS), 4′,6-diamidino-2-phenylindole (1:1000; FUJIFILM Wako Pure Chemical, Osaka, Japan) and the following secondary antibodies were applied: Cy3-conjugated anti-mouse IgG (1:200; Jackson ImmunoResearch Laboratories, West Grove, PA, USA) and Alexa Fluor-488-conjugated anti-rabbit IgG (1:200; Jackson ImmunoResearch Laboratories). The tissue samples were incubated at 25 °C for 1 h. The samples were thoroughly washed thrice with PBS for 5 min before mounting. Slides were visualized using a confocal laser-scanning microscope (LSM700; Carl Zeiss, Oberkochen, Germany) and analyzed using ZEN 2009 (Carl Zeiss).

### 2.4. Single-Cell RNA Sequence (scRNA-seq)

The analyzed scRNA-seq dataset of 6 patient-matched colorectal normal and CRC tissues (GSE144735) [18] was downloaded and reanalyzed using R version 4.3.3 with the Seurat R package v.3.0.0. In the dataset, 1 of pT1, 2 of pT4a, and 3 of pT3 CRCs were included among a total of 6 CRCs. “Normal” and “Border” (a sample from the peripheral lesion of a CRC) class cells were selected for gene expression change analysis. Further, “Myofibroblast” and “Stromal 1” clusters of “Border” class cells were re-clustered using the following procedure. The NormalizeData, FindvariableFeatures, and RunPCA functions were used to determine significant principal components. Twenty components were retained based on the ElbowPlot data, and the FindNeighbors and FindClusters functions were applied. Finally, dimension reduction was performed using the RunUMAP function, and new clustering was generated.

### 2.5. The Cancer Genome Atlas (TCGA) Database

TCGA data of colon and rectal adenocarcinoma (TCGA, GDC) are accessible on the cBioportal (https://www.cbioportal.org/ (accessed on 21 September 2024)). Major mutated genes with their mutation rates were searched.

### 2.6. Statistical Analyses

For comparison of PDPN expression by IHC, SPSS Statistics ver. 22 (IBM, Chicago, IL, USA) was used to test the statistical significance. Maximum PDPN expression in the LP was compared using Pearson’s chi-square test for unpaired data and the Wilcoxon signed-rank test for case-paired data. Pericryptal PDPN expression was compared using the Kruskal–Wallis test. Receiver operating characteristic (ROC) curves were used to evaluate the diagnostic value of pericryptal PDPN expression. For some cutoff values derived from the ROC analysis, sensitivity, specificity, and positive or negative predictive values (PPV or NPV, respectively) were calculated manually. The ggpubr package in R was used to analyze scRNA-seq data. Statistical significance was set at *p* < 0.05.

## 3. Results

### 3.1. Stroma of Early AC Exhibits High PDPN Expression, Lowering When the AC Invades Deeper

In early AC, the LP exhibited higher PDPN expression than the normal adjacent mucosa, with a clear demarcation line between them (Figure 1A,B). Conversely, the lumen-side stroma expressed PDPN, whereas the stroma at the original muscularis propria or subserosa/adventitia levels had much lower PDPN expression in advanced AC (Figure 1C). This trend was observed in early AC. As AC glands invaded and reached the submucosal level, stromal PDPN expression disappeared or attenuated. However, some glands retained the PDPN lining, which could be mistaken for LI. Lymphatic vessels exhibited the strongest PDPN expression in colorectal PDPN IHC specimens and had smooth, continuous, and monolayered shapes (Figure 1D). This differentiation helped distinguish the periglandular PDPN lining of AC glands from lymphatic vessels. These observations led us to hypothesize that PDPN-expressing stromal cells appear early in colorectal carcinogenesis and differ from stromal cells forming in advanced AC.

### 3.2. Stroma of Early AC or Adenoma and Inflamed LP Express Higher Levels of PDPN than NET Stroma, MPS, and Normal LP

We evaluated stromal PDPN expression in the LP as either negative or positive. Positive cases were further subdivided into low- and high-expression groups based on the PDPN immunoreactivity of the LECs (Figure 2). All glandular tumor cases (early AC and adenoma) exhibited PDPN expression in the LP, with the majority showing high expression (71/73 in early ACs and 35/39 in adenomas). Conversely, low or negative expression was more frequently observed in their normal adjacent mucosae. NET stroma showed slightly higher PDPN expression than its non-neoplastic adjacent mucosa; however, the difference was not statistically significant. Many inflamed mucosae were positive for PDPN, with a tendency toward high intensity (20/36 inflamed mucosae). MPS cases showed a similar pattern to the adjacent mucosa of adenomas (Table 2). In diseased LP, two distinct PDPN expression patterns were observed: (1) PDPN-positive cells scattered or forming a reticular network in the stroma (reticular pattern) and (2) PDPN-positive cells lining crypts (pericryptal pattern) (Figure 2, PDPN (high-power)). Small, round, or spindle-shaped cells in the LP expressed PDPN. Their cytoplasm occasionally formed a reticular network corresponding to the type-III collagen network highlighted with reticulin staining (Appendix A), suggesting that these stromal cells were RCs. Although RCs expressed PDPN in their normal state, their expression was enhanced in diseased areas. Many tumor glands (early AC and adenoma) were lined with linear and continuous PDPN-positive cells (Figure 1 and Figure 2). This crypt lining resembled that of pericryptal fibroblasts (PCFs), a unique myofibroblast syncytium expressing αSMA.

### 3.3. Pericryptal Fibroblasts May Change into PDPN-Positive “Early-Stage CAFs”

On αSMA IHC, non-neoplastic crypts were robustly surrounded by αSMA-positive PCFs, whereas some AC glands lacked αSMA lining as previously reported [19], with variation from gland to gland. Notably, αSMA-negative LP areas were also observed. This suggested that the loss of pericryptal αSMA lining was not solely from detachment and migration of PCFs (Appendix A). Double immunofluorescence of PDPN and αSMA demonstrated non-neoplastic crypts lined predominantly by αSMA, whereas AC glands often lacked αSMA lining and acquired PDPN lining. Some pericryptal cells co-expressed PDPN and αSMA (Figure 3A). Subsequently, we reanalyzed publicly available scRNA-seq data (GSE144735) [18] using the Seurat package. Stromal cells had already been subdivided into myofibroblasts, pericytes, and other cell types (Figure 3B). Stroma in the peripheral region of CRC (“Border”) was more abundant in *PDPN* (the gene encodes PDPN protein) than in non-neoplastic adjacent colorectal tissue (“Normal”) (Figure 3C,E and Appendix A). CAF marker genes including *FAP*, *POSTN*, *ACTA2* (the genes encode FAP, POSTN, and αSMA proteins, respectively), and *PDPN* were highly expressed in myofibroblasts and stromal 1 cells; however, the distribution of *PDPN* and the other three genes did not overlap (Figure 3D). When the cluster map was split into “Normal” and “Border” class cells, many myofibroblasts and stromal 1 cells were found only in the “Border” class (Figure 3E and Appendix A). We subsequently created a subset composed of myofibroblasts and stromal 1 cells and defined them as CAFs. CAFs were further analyzed and divided into seven clusters (clusters 0–6, Figure 3F). The clusters were separated into *PDPN*^high^ CAFs (clusters 0 and 5); *ACTA2*, *FAP*, and *POSTN*^high^ CAFs (clusters 1 and 2); and miscellaneous CAFs (clusters 3, 4, and 6). The former CAFs also highly expressed *COL4A1*, *COL4A2*, and *WNT5A* (Figure 3G and Appendix A).

### 3.4. Pericryptal PDPN Expression Is Indicative of Glandular Tumors

Given that whether PDPN was negative or positive (both low and high) in the LP did not clearly distinguish between glandular tumors and inflamed mucosae, we focused on the pericryptal PDPN expression pattern. Many tumor glands (early AC and adenoma) were lined with linear and continuous PDPN-positive cells (Figure 2). This crypt lining resembled the lining by αSMA-positive PCFs, and we counted “pericryptal positivity” of PDPN according to a previous report on PCFs with minor modification (Figure 4A) [16]. Notably, glandular tumors showed significantly higher pericryptal PDPN positivity than did non-neoplastic lesions (non-neoplastic adjacent mucosa (NNA), MPS, and inflammation) (Figure 4B). To test the ability of pericryptal positivity to distinguish non-neoplastic lesions from glandular tumors, ROC curve analysis was performed. The area under the curve (AUC) was 0.960, and the theoretical best pericryptal positivity cutoff value on the ROC curve was 3.72% (Figure 4C). When a cutoff of 3.72% was used, the sensitivity was 93.7%; specificity, 91.2%; PPV, 87.4%; and NPV, 95.7%. However, when the 10% cutoff was used, the sensitivity was 87.4%; specificity, 97.1%; PPV, 95.1%; and NPV, 92.2% (Appendix A).

### 3.5. Pericryptal PDPN Expression Better Detects AC and Adenoma at Their Early Stage than p53 Aberrant Expression on IHC

To compare PDPN’s ability to detect early colorectal tumors, p53 IHC was performed as an established marker (Appendix A). Inflammation, MPS, and non-neoplastic adjacent mucosa of adenoma or early AC exhibited wild-type p53 expression. A small proportion of adenomas (5/39, 12.8%) showed aberrant p53 expression, whereas nearly 70% of early AC (45/67, 67.2%) showed aberrant p53 expression, corresponding to reported *TP53* mutation rates (Table 3). As shown in Table 3, p53 aberrant expression had a sensitivity of 47.2%; specificity, 100%; PPV, 100%; and NPV, 74.6% for glandular tumors (early AC and adenoma) (Appendix A). In many early AC and adenoma cases, cells with p53 aberrant expression pattern emerged within neoplastic cells with p53 wild-type pattern (Appendix A). No significant correlation between pericryptal PDPN positivity and p53 aberrant expression was observed (Table 4).

## 4. Discussion

Here, we demonstrated the significance of PDPN as an early-stage CAF marker in CRC and the diagnostic value of PDPN-positive PCFs in colorectal lesions. Early ACs and adenomas have higher pericryptal PDPN expression than their normal adjacent counterparts [13]. The present study focused on PDPN expression patterns in multiple types of colorectal lesions and investigated the differences in expression patterns. A consistent LP reticular pattern was observed across all lesion types, whereas the pericryptal lining pattern was dominantly observed in early ACs and adenomas. We hypothesized that PCFs in glandular tumors express PDPN. PCFs represent a unique αSMA-positive myofibroblast syncytium surrounding gastrointestinal glands. Crypts/glands of AC and adenoma exhibit fewer αSMA linings than normal crypts, suggesting a reduction or disappearance of PCFs as the tumor progresses [16,19]. In contrast, Kaye et al. conducted a study on adenoma glands using electron microscopy and revealed a failure of morphological maturation and collagen production in PCFs. They also demonstrated that PCFs were attached to the crypts at all levels [20]. Our IHC findings demonstrated that not all AC crypts lacked αSMA surrounding, and αSMA and PDPN double-positive PCFs were found in early AC double immunofluorescence. We propose that PCFs “remain” even after carcinogenesis begins and “change” their characteristics, becoming αSMA^low^ and PDPN^high^. In non-neoplastic lesions, various inflammatory stimuli in the LP may induce PDPN, primarily in RCs and secondarily in PCFs. Tumor cells instruct stromal cells on CAFs via indirect and direct interaction [7,8,10]. We hypothesize that PDPN expression in PCFs of glandular tumors is primarily induced by tumor cell–PCF interactions. RCs and some normal adjacent mucosae of ACs and adenomas expressed PDPN, probably because of PDPN induction by humoral factors secreted by tumor cells. PCFs are known to radiate into the LP to form a fibroblastic syncytium throughout the colorectal mucosa [21,22]. Therefore, PDPN expression may be shared by PCFs and RCs under both non-neoplastic and neoplastic conditions. In both early and advanced ACs, PDPN expression decreases as AC invades deeper. Thus, we hypothesized that PCFs and RCs are sources of early-stage CAFs in CRC and that CAFs present in advanced AC have different cellular origins, functions, and marker gene expression.

To investigate CAF marker distribution, we used a scRNA-seq dataset composed of normal colorectal mucosae and CRCs. As anticipated, *PDPN*^high^ CAFs differed from *ACTA2*, *POSTN*, and *FAP*^high^ CAFs. These three factors are established CAF markers in advanced CRC [23,24,25]. Nishishita et al. studied PDPN, αSMA, platelet-derived growth factor receptor-β (PDGFR-β), and type-I collagen expression of advanced CRC stroma using IHC. Their findings revealed that PDPN expression did not correlate with lymphatic or venous invasion, lymph node metastasis, αSMA, PDGFR-β, and type-I collagen expression. Similar to our advanced AC case, PDPN expression was localized in the luminal side stroma [23]. This supports our hypothesis that PDPN-positive CAFs are derived from superficial fibroblasts, including PCFs and RCs. To confirm our hypothesis, we needed to identify a PCF marker other than αSMA, as its expression decreases in PCFs during carcinogenesis. PCFs attach to the basement membrane of crypts and are thus likely involved in the production or maintenance of the basement membrane. An in vitro experiment demonstrated enhanced production of type-IV collagen by 36T-2 fibroblasts upon direct contact with DLD-1 colonic epithelial cells, suggesting that PCFs in close proximity to epithelial cells contribute to type-IV collagen synthesis [26]. The basement membrane containing type-IV collagen is well preserved in the superficial portion of CRC; however, it diminishes as the cancer invades deeper into the mucosa, mirroring the pattern of pericryptal/periglandular PDPN expression [27]. These observations suggested that PCFs may express genes encoding type-IV collagen. In the present scRNA-seq analysis, CAF clusters 0 and 5 exhibited high expression of *PDPN*, *COL4A1*, and *COL4A2* and low expression of *ACTA2*, *POSTN*, and *FAP*. *COL1A1* and *COL1A2* expression was also lower in these clusters than in clusters 1 and 2, which exhibited high *ACTA2*, *POSTN*, and *FAP* expression.

Wnt family members and platelet-derived growth factor receptor alpha (PDGFR-α) are known as PCF markers [28]. *PDGFRA* expression was dominant in clusters 0, 3, and 5. Notably, *WNT5A* exhibited high accumulation in clusters 0 and 5. Wnt5a-positive PCFs are typically located at the top level of a crypt in the normal state and increase in number during mucosal injury or inflammation, serving as an activated PCF marker. From these observations, we posit that clusters 0 and 5, which are *PDPN*^high^ CAFs, may represent superficial early-stage CAFs derived from PCFs. Although the expression of typical RC marker genes, including *CCL19* and *MADCAM1* [29], was low in the CAF clusters, we were unable to definitively determine whether RCs served as a source of podCAFs. However, given that PCFs and RCs form syncytia, they cannot be completely separated. Other types of colorectal lesions, including serrated lesions and idiopathic inflammatory bowel disease, warrant future investigation. NET, while tumors, exhibited stroma with low levels of PDPN-poor stroma, indicating distinct microenvironments from glandular tumors. Our results suggest that PDPN in PCFs may serve as a novel marker for glandular tumors in diagnostic pathology. Colorectal epithelial cells often display cytological and structural atypia during inflammation and regeneration, with MPS being a good example that may mimic a neoplasm [30]. Pathologists occasionally face challenges in determining whether lesions are neoplastic based solely on morphological characteristics of the epithelium. According to The Cancer Genome Atlas (TCGA) data, *APC* or *TP53* gene alterations are most frequently found in CRC (74.2% and 59.6%, respectively). Theoretically, detecting such genomic abnormalities using IHC is powerful evidence of neoplasm, but, unfortunately, IHC methods that reliably detect *APC* mutation have not been developed yet. APC is involved in the degradation of β-catenin and is usually bound to E-cadherin on the plasma membrane. When APC’s normal function is lost, β-catenin accumulates in the cytoplasm and, subsequently, in the nucleus. However, Brabletz et al. reported that even in *APC*-mutated CRC cells, β-catenin was found along the plasma membrane when the cells were attaching to other cells or were forming a glandular structure [31]. Awad et al. described that cytoplasmic and intranuclear β-catenin expression were found only in 69.8% and 36.5% of CRC cases using IHC, respectively [32]. Based on such reports, β-catenin IHC cannot act as a surrogate marker of *APC* mutation.

However, p53 IHC well reflects *TP53* status. Fearon et al. described that in the adenoma–carcinoma sequence of sporadic CRC, *TP53* alteration usually occurred in late-stage adenoma [33]. This means that many adenomas have a normal *TP53* gene and exhibit wild-type p53 expression on IHC, as supported by our present data. Kim et al. reported that approximately 70% of CRCs harbored *TP53* variation, and a nearly equal proportion of cases exhibited aberrant p53 expression on IHC. Notably, more advanced CRC showed higher TP53 variation and p53 aberrant expression rates (66.4% in stage I and II vs. 80.9% in stage III and IV and 66.4% in stages I and II vs. 83% in stages III and IV, respectively) [34]. In our case series, p53 aberrant expression was detected in 67.2% of early AC, corresponding to Kim’s report. Only 12.8% of adenoma cases had p53 aberrant expression. Notably, the aberrant expression was often observed only in a part of the early AC or adenoma lesion, indicating that a *TP53*-mutated clone emerged after adenoma or AC development. This makes detection of *TP53* mutation difficult because performing genomic analysis or IHC for the entire lesion is costly and time-consuming. Collectively, p53 IHC is not good at detecting early-stage colorectal neoplasms, and it has limited sensitivity even for advanced CRC. In contrast, the emergence of PDPN-positive PCFs strongly indicates tumorigenesis, with higher pericryptal PDPN positivity in early AC and adenoma cases supporting this notion. Pericryptal PDPN positivity exhibited excellent diagnostic power (AUC = 0.960), and we consider this model as practical in daily practice. Upon comparing the cutoff values of 3.72% and 10%, 10% had high and well-balanced PPV, and NPV and had very high specificity (97.1%). Thus, we propose the 10% cutoff to be better because immunohistochemistry is not a screening test and requires high specificity. Additionally, 10% is a round value and easy to use. Pathologists should be aware that PDPN is not specific to LECs. LI is a crucial risk factor of lymph node metastasis, with additional surgery needed if LI is present in endoscopically resected samples. Therefore, misdiagnosing periglandular PDPN lining as LI must be avoided to prevent overtreatment.

This study has some limitations. First, the function of podCAFs in early AC has not yet been determined. We have proposed that PDPN is an early-stage CAF marker; however, we did not investigate whether podCAFs promote or suppress cancer progression. Considering a previous report on podCAFs in CRCs [12], they may function as barriers to cancer invasion. Experimental studies, such as co-culture and animal models, are needed to elucidate their functions. Second, sources of advanced-stage CAFs were not thoroughly investigated in this study as our focus was on PDPN and PCFs. We could not determine whether early-stage CAFs (PDPN^high^) transformed into advanced-stage CAFs (αSMA, POSTN, or FAP^high^) or if other cell types, such as smooth muscle cells [35], bone marrow-derived stem cells [5], or adipocytes [6], acquired CAF characteristics. Kobayashi et al. reported that melanoma cell adhesion molecule (MCAM)-positive PCFs proliferated and changed into αSMA-positive CAFs [36]. Although evidence for MCAM-positive stromal cells was lacking, except for pericyte clusters in the scRNA-seq, a subpopulation of PCFs may still contribute to the generation of advanced-stage CAFs. Finally, we used the scRNA-seq dataset from advanced CRC cases for our analysis. Transcriptomic analysis requires an abundant number of cells; however, early CRCs resected using endoscopic methods are typically small and thin. Pierced or ulcerated tissue after sampling is unsuitable for precise histological diagnosis. To overcome this challenge, methods that preserve sample integrity, such as spatial transcriptome analysis, may be advantageous.

## 5. Conclusions

Our study revealed that RCs and PCFs acquire PDPN expression during colorectal carcinogenesis, changing into early-stage CAFs. As AC invades deeper, the CAFs surrounding the carcinoma cells become dominated by PDPN-negative cells, representing advanced-stage CAFs. This dynamic CAF transition highlights the complexity and plasticity of the cancer microenvironment (Figure 5). Understanding both advanced- and early-stage CRC microenvironments is crucial for comprehending CRC pathobiology and achieving early detection and treatment of CRC in the future.

## Figures and Tables

**Figure 1 cells-13-01682-f001:**
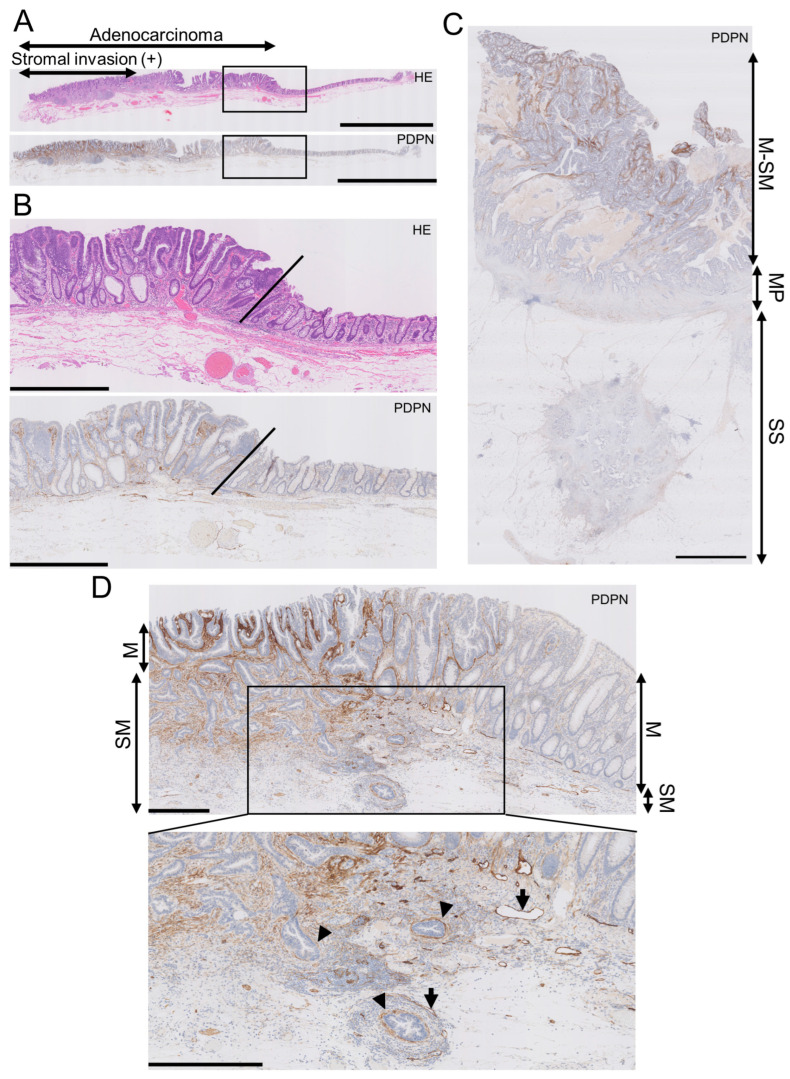
Podoplanin (PDPN) is highly expressed in the lamina propria mucosa of early AC, and the expression lowers as AC invades the deeper portion. (**A**) Low-power view of an early AC resected via endoscopic submucosal dissection (ESD). Scale bar = 5 mm. (**B**) High-power view of the AC is shown as rectangles in A. Oblique lines indicate the normal-neoplasm front. Scale bar = 1 mm. (**C**) PDPN IHC of advanced AC. PDPN expression decreases as the AC invades the bowel wall. Scale bar = 2.5 mm. (**D**) Low-power (**upper**) and high-power (**lower**) views of early AC resected via ESD. Even in early AC, PDPN attenuation occurs in the invasive portion. Arrowheads indicate pericryptal/periglandular PDPN linings. Arrows indicate lymphatic vessels. Scale bar = 500 µm. HE, hematoxylin and eosin staining; M, mucosa; SM, submucosa; MP, muscularis propria; SS, submucosa.

**Figure 2 cells-13-01682-f002:**
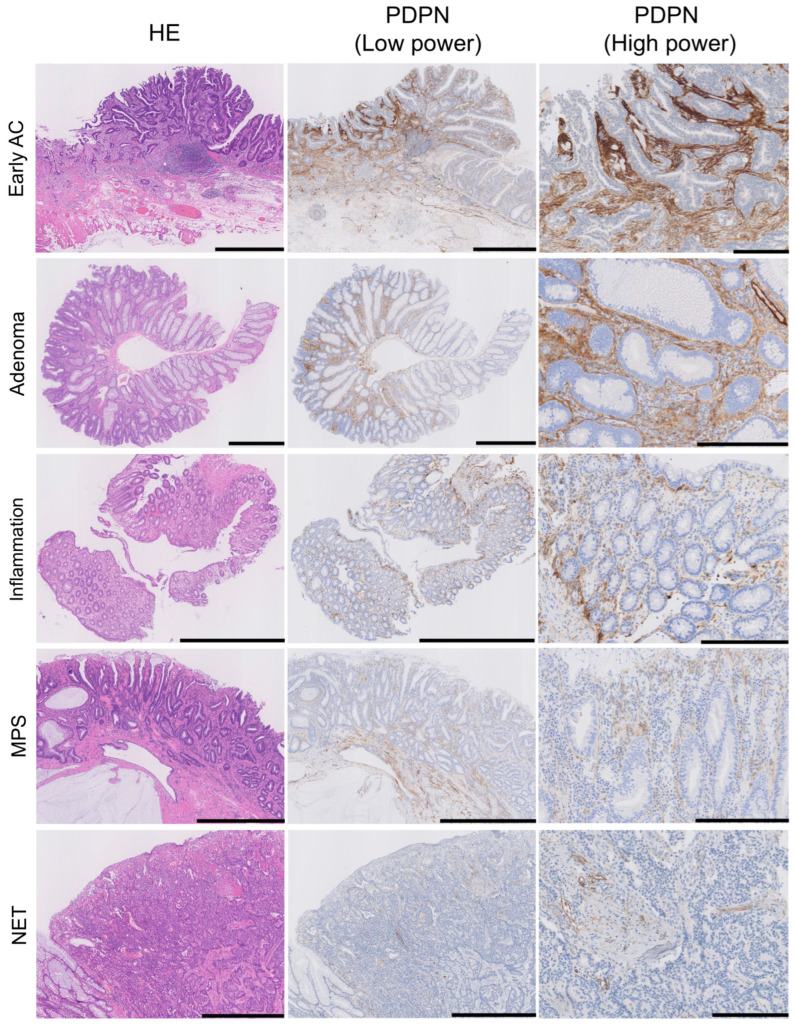
PDPN is also expressed in the lamina propria mucosae of adenomas, inflamed mucosa, and MPS or in NET stroma. HE, low-power PDPN IHC, and high-power PDPN IHC images are arranged from the left where PDPN expression is observed in non-neoplastic lesions; the intensity is higher in AC and adenoma. Pericryptal PDPN expression is predominant in early AC and adenoma, whereas reticular expression in the LP was observed in all lesions. Scale bar = 1 mm for HE and PDPN (low power) and 250 µm for PDPN (high power). HE, hematoxylin and eosin staining; AC, adenocarcinoma; MPS, mucosal prolapse syndrome; NET, neuroendocrine tumor.

**Figure 3 cells-13-01682-f003:**
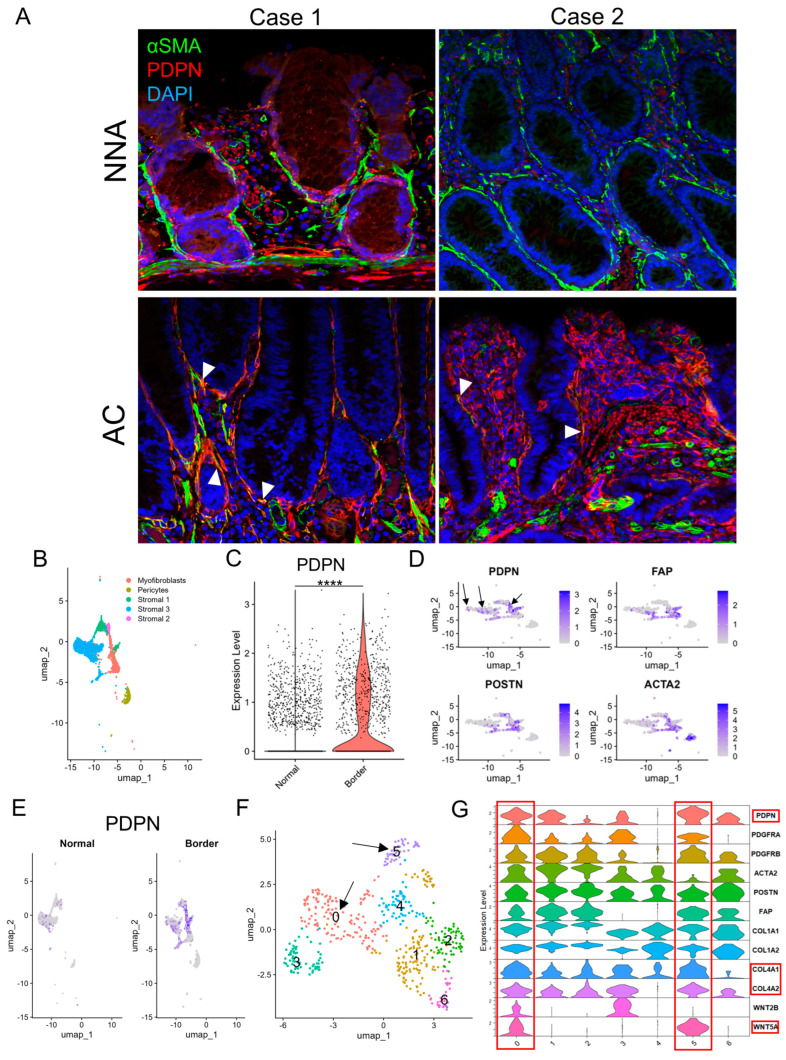
PDPN is likely induced in pericryptal fibroblasts during glandular tumorigenesis. (**A**) Double immunofluorescence staining of αSMA and PDPN on early ACs and their non−neoplastic adjacent mucosae. Case 1 represents a colonic mucosa with high PDPN expression in a non−neoplastic state. Case 2 represents weak PDPN expression in the non-neoplastic mucosa. Arrowheads indicate co−expression of αSMA and PDPN. (**B**–**F**) scRNA−seq of the GSE144735 dataset. (**B**) UMAP plot of stromal cells of “Normal” and “Border” classes, excluding endothelial cells. (**C**) Violin plot comparing PDPN expression in fibroblasts of normal colorectal mucosa (Normal) and CRC (Border). (**D**) Feature plots of CAF marker genes (*PDPN*, *FAP*, *POSTN*, and *ACTA2*) on the cluster map of stromal cells shown in B. Arrows indicate *PDPN*−positive cells with no or few overlaps with *FAP*, *POSTN*, or *ACTA2* expression. (**E**) PDPN feature plot on the cluster map shown in B split into “Normal” and “Border” class cells. (**F**) UMAP plot of re-clustered myofibroblasts and stromal 1 cells. These cells were newly subdivided into clusters 0−6. Arrows point to clusters 0 and 5 with high PDPN accumulation as shown in (**G**). (**G**) Violin plot of selected CAF and PCF marker genes for fibroblasts re-clustered in F. Clusters 0 and 5 are rich in PDPN, *COL4A1*, *COL4A2*, and *WNT5A* expression and highlighted with red rectangles. PDPN, podoplanin; NNA, non-neoplastic adjacent mucosa; AC, adenocarcinoma; UMAP, uniform manifold approximation and projection. ****: *p* < 0.0001.

**Figure 4 cells-13-01682-f004:**
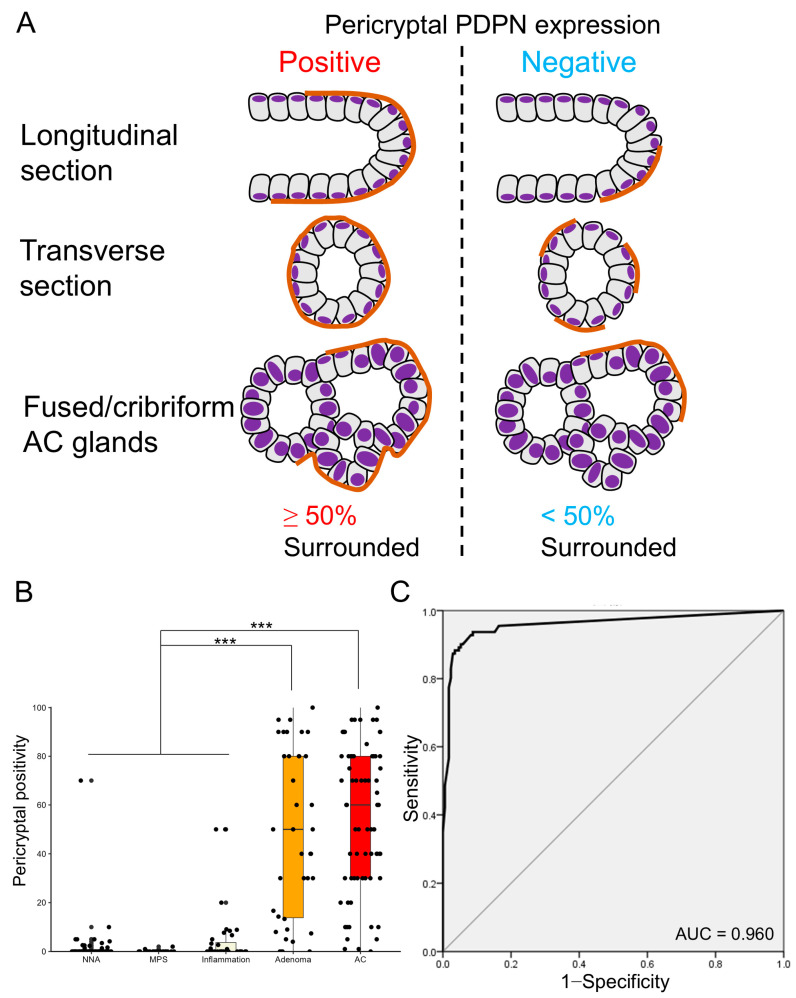
Pericryptal PDPN expression is more prominent in early AC and adenoma. (**A**) Definition of pericryptal PDPN expression in this study. Similar to a previous study on PCFs, we define glands with ≥50% of continuous lining by PDPN-expressing cells as positive for pericryptal expression. Fused or cribriform AC glands are defined as a single gland for the evaluation of pericryptal PDPN expression. (**B**) Boxplot and jitter plots of pericryptal PDPN positivity. Adenomas and early ACs show significantly higher pericryptal positivity than NNA, MPS, or inflammation. Data are analyzed using the Kruskal–Wallis test with SPSS software ver. 22. NNA, non-neoplastic adjacent mucosa; MPS, mucosal prolapse syndrome; AC, adenocarcinoma; *** *p* < 0.001. (**C**) Receiver operating characteristic (ROC) curve evaluating the diagnostic power of pericryptal positivity in distinguishing non-neoplastic lesions (NNA, MPS, and inflammation) from glandular tumors (adenoma and early AC). HE, hematoxylin and eosin staining; AC, adenocarcinoma; AUC, area under the curve.

**Figure 5 cells-13-01682-f005:**
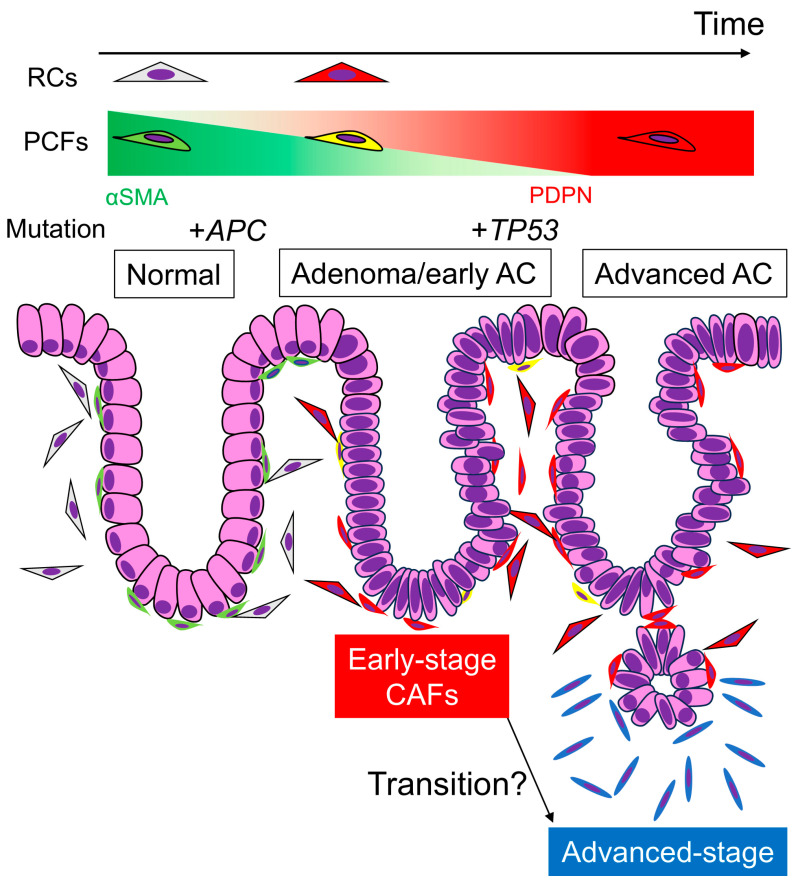
Transition of PDPN expression in stromal fibroblasts and representative gene mutations via colorectal tumorigenesis and carcinoma progression. Under normal conditions, RCs and PCFs show little or no PDPN expression. Once adenomas develop, these fibroblasts acquire PDPN expression and lose αSMA expression to be “early-stage CAFs”, and the expression pattern remains after the adenoma transforms into early AC. *APC* mutation occurs at the beginning of colorectal tumorigenesis, which is undetectable by IHC. *TP53* mutation occurs when or after the adenoma transforms into AC. When the AC invades the submucosa or deeper portions, PDPN expression in stromal CAFs diminishes. These “advanced-stage CAFs” likely express other CAF markers, such as αSMA or FAP. It remains unclear whether “early-stage CAFs” derived from PCFs or RCs transform into “advanced-stage CAFs.” Perhaps the latter CAFs have completely different origins, including adipocytes and bone marrow-derived mesenchymal stem cells. AC, adenocarcinoma; RCs, reticular cells; PCFs, pericryptal fibroblasts; CAFs, cancer-associated fibroblasts.

**Table 1 cells-13-01682-t001:** Demographic and clinicopathological characteristics of samples.

	Early AC (n = 73) ^b^	Adenoma (n = 39)	NET (n = 11)	Inflammation (n = 36) ^c^	MPS (n = 27)
Sex					
Male	48	28	7	10	16
Female	22	11	4	9	11
Method					
ESD	68	0	11	0	1
EMR	5	7	0	0	0
CSP	0	32	0	0	4
Biopsy	0	0	0	36	22
Location ^a^					
Cecum	10	3	0	7	0
Ascending colon	11	10	0	4	0
Transverse colon	12	11	0	7	0
Descending colon	5	4	0	4	0
Sigmoid colon	13	9	0	6	0
Rectosigmoid	4	1	0	1	0
Rectum, part unspecified	0	1	0	4	7
Upper rectum	4	0	2	0	0
Lower rectum	14	0	9	3	20
pT (Early AC) ^a^					
is	8				
1a	30				
1b	35				
pT (NET) ^a^					
1a			9		
1b			2		
Lymphatic invasion ^a^					
Negative	61		7		
Positive	12		4		
Venous invasion ^a^					
Negative	62		9		
Positive	11		2		
Tumor budding ^a^					
BD1	59				
BD2	4				
BD3	2				

AC, adenocarcinoma; NET, neuroendocrine tumor; MPS, mucosal prolapse syndrome; ESD, endoscopic submucosal dissection; EMR, endoscopic mucosal resection; CSP, cold snare polypectomy; BD, budding. ^a^ According to the Japanese Classification of Colorectal Cancer, 9th edition [10]. For early AC, pTis, carcinoma confined to the mucosa; pT1a, carcinoma confined to the submucosa and invasion < 1000 µm; pT1b, carcinoma confined to the submucosa and invasion ≥ 1000 µm. For NET, pT1a, tumor size < 1 cm; pT1b, 1 cm ≤ tumor size ≤ 2 cm. BD1, 0–4 tumor bud (s) on ×200 field; BD2, 5–9 tumor buds on ×200 field; BD3, 10 or more tumor buds on ×200 field. ^b^ 73 samples from 70 patients. ^c^ 36 samples from 19 patients.

**Table 2 cells-13-01682-t002:** Maximum PDPN expression in the lesion with reference to that in lymphatic endothelial cells.

	Maximum PDPN Expression ^a^				
	Negative (%)	Low (%)	High (%)	*p*-Value (vs. Early AC)	*p*-Value (vs. Adenoma)
Early AC	0 (0)	2 (2.7)	71 (97.3)	—	0.92 ^c^
Early AC NNA	16 (24.2)	25 (37.9)	25 (37.9)	<0.001 ^b^	—
Adenoma	0 (0)	4 (10.2)	35 (89.8)	0.92 ^c^	—
Adenoma NNA	11 (28.2)	20 (51.3)	8 (20.5)	—	<0.001 ^b^
Inflammation	7 (19.4)	9 (25)	20 (55.6)	<0.001 ^c^	0.002 ^c^
MPS	7 (25.9)	16 (59.3)	4 (14.8)	<0.001 ^c^	<0.001 ^c^
NET	6 (54.5)	3 (27.3)	2 (18.2)	<0.001 ^c^	<0.001 ^c^
NET NNA	8 (72.7)	3 (27.3)	0 (0)	<0.001 ^c^	<0.001 ^c^

AC, adenocarcinoma; NNA, non-neoplastic adjacent tissue; MPS, mucosal prolapse syndrome; NET, neuroendocrine tumor. ^a^ Negative, no expression; Low, weaker than LECs; High, equal to or stronger than LECs. ^b^ Data are analyzed with Wilcoxon signed-rank test. ^c^ Data are analyzed with Pearson’s chi-square test.

**Table 3 cells-13-01682-t003:** p53 expression on IHC for selected samples.

	p53 Expression Pattern (%)		Total
	Wild-Type	Aberrant	
Early AC	22 (32.8)	45 (67.2)	67
Early AC NNA	62 (100)	0 (0)	62
Adenoma	34 (87.2)	5 (12.8)	39
Adenoma NNA	39 (100)	0 (0)	39
Inflammation	36 (100)	0 (0)	36
MPS	27 (100)	0 (0)	27

Percentage of p53 aberrant expression in early AC and adenoma indicates sensitivity. AC, adenocarcinoma; NNA, non-neoplastic adjacent tissue; MPS, mucosal prolapse syndrome.

**Table 4 cells-13-01682-t004:** Relationship between pericryptal PDPN positivity and p53 expression of tumor cells.

			p53 Expression Pattern		*p*-Value
			Wild-Type	Aberrant	
Early AC	Pericryptal positivity	<10%	1	4	0.525
		≥10%	21	41	
Adenoma		<10%	8	1	0.861
		≥10%	26	4	

AC, adenocarcinoma. Data are analyzed with Pearson’s chi-square test.

## Data Availability

The data presented in the present study are available on request from the corresponding author (S.T.).

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
