# Peer review of "Podoplanin Expression in Early-Stage Colorectal Cancer-Associated Fibroblasts and Its Utility as a Diagnostic Marker for Colorectal Lesions"

_cells, 2024, doi:10.3390/cells13201682_

Round 1

Reviewer 1 Report (Previous Reviewer 2)

Comments and Suggestions for Authors

I am happy that the authors have addressed my concerns fully (although I think the language needs tidying up before being considered for publication) and therefore would be willing to recommend publication.

Comments on the Quality of English Language

I think the language needs tidying up before being considered for publication

Reviewer 2 Report (New Reviewer)

Comments and Suggestions for Authors

The manuscript presents an original and thorough investigation on cancer-associated fibroblasts (CAFs’) phenotypes in different types and stages of colorectal lesions.

The Introduction is well focused and structured and based on CAFs, their markers and on the diagnostic difficulties related to distinguishing non-neoplastic from neoplastic colorectal lesions. The focus on podoplanin (PDPN) expression.

The Materials and Methods section is presented in detail and the methodology is easily reproducible. A variety of informative and adequate to the aim of the study assays have been used. My only comment is on the scale of IHC assessment of PDPN  staining intensity. The authors state that the expression is classified as high or low compared to that of LECs. Normally, a quantitative or at least a semiquantitative scale is used (if no specialized software) assessing the number/percentage of positive cells.

The Results are illustrated by 5 highly informative figures and 3 tables. The composite figures combine immunofluorescence, scRNA-seq, feature plots of CAF marker genes or graphic presentations with boxplot and jitter plots of pericryptal PDPN positivity and ROC curves. The results are presented concisely. They show that pericryptal fibroblasts and reticular cells in lamina propria are percurcors of early-stage CRC CAFs. RCs and PCFs acquire PDPN expression in the course of colorectal malignant transformation, changing into early-stage CAFs. As the process advances deeper, peritumor CAFs change their phenotype to  PDPN-negative cells. These cells are considered as advanced-stage CAFs.

The Discussion section is analytical and comprehensive. The limitations of the study could are correctly defined.

The Conclusions are based on the results obtained. 

The English language and the overall style of the manuscript need no linguistic revision. The references are relevant.

In conclusion, the manuscript offers a detailed study on the dynamic CAF transition during colorectal carcinogenesis based on the investigation of several marker proteins and genes.

The manuscript is suitable for publication. 

This manuscript is a resubmission of an earlier submission. The following is a list of the peer review reports and author responses from that submission.

Round 1

Reviewer 1 Report

Comments and Suggestions for Authors

The authors present a manuscript that looks at Podoplanin expression in early-stage colorectal cancer-associated fibroblasts and whether this can be used diagnostically. I think this manuscript has some significant results and should be publishable. I have a few minor comments.

In the abstract I think it would be beneficial to expand slightly on the results section and describe better the main findings. 

The introduction I thought was fairly well written and concise - It may be interesting if the authors summarise in slightly more detail reference 13 which identified this phenomena previously. 

I think there is also scope to expand slightly on the specific objectives at the end of the introduction.

The methodology section

Can the authors explain the differences in Male/Female ratio for this study design and how this was accounted for in analysis of results.

How was the histological diagnostic criteria established for this study i.e. it involves mutliple clinicial personnel and giving averaged scores etc. How did you decide on agreement of <20% etc.

Results

I think generally the results are well presented but some figures could be better annotated to explain the main features to the reader. 

At the end of the results/discussion iut would be good to summarise the main stages to making this a diagnostic test and what are the barriers i.e. the ROC curve looks good but does this reflect PPV/NPV overall accuracy of diagnosis in a clinical setting?

Comments on the Quality of English Language

English was fine from my perspective

Author Response

Authors appreciate your valuable review. Please see the attachment for our responses.

Reviewer 2 Report

Comments and Suggestions for Authors

A fairly comprehensive study proposing podoplanin as an early detection diagnostic marker for CRC, with a good number of neoplastic and non-neoplastic lesions included.

To strengthen the claims for podoplanin, I think the study would be greatly improved by also comparing in the main IHC study against other potential early detection markers, both related to CAF and colon epithelial-cells.

Comments on the Quality of English Language

Good - only a few typographical errors.

Author Response

(The authors gave the same response as above.)

Round 2

Reviewer 2 Report

Comments and Suggestions for Authors

I still feel the study needs strengthening as I suggested previously. 

Comments on the Quality of English Language

Quality is good

Author Response

Thank your for your review again. Please see the attachment for our response.
